# Intraperitoneally Delivered Umbilical Cord Lining Mesenchymal Stromal Cells Improve Survival and Kidney Function in Murine Lupus via Myeloid Pathway Targeting

**DOI:** 10.3390/ijms24010365

**Published:** 2022-12-26

**Authors:** Alvin Wen Choong Chua, Dianyang Guo, Jia Chi Tan, Frances Ting Wei Lim, Chee Tian Ong, Jeyakumar Masilamani, Tony Kiat Hon Lim, William Ying Khee Hwang, Ivor Jiun Lim, Jinmiao Chen, Toan Thang Phan, Xiubo Fan

**Affiliations:** 1Department of Plastic, Reconstructive and Aesthetic Surgery, Singapore General Hospital, Singapore 169856, Singapore; alvin.chua.w.c@singhealth.com.sg; 2Department of Clinical Translational Research, Singapore General Hospital, Singapore 169608, Singapore; guo.dianyang@sgh.com.sg (D.G.); frances.lim.t.w@sgh.com.sg (F.T.W.L.); 3Single-Cell Computational Immunology, Singapore Immunology Network, Singapore 138668, Singapore; e0546189@u.nus.edu (J.C.T.); chen_jinmiao@immunol.a-star.edu.sg (J.C.); 4CellResearch Corporation Pte Ltd., Singapore 048943, Singapore; ongcheetian@cordlabs.sg (C.T.O.); jeyakumar@cordlabs.sg (J.M.); ivorlim@cellresearchcorp.com (I.J.L.); 5Department of Anatomical Pathology, Singapore General Hospital, Singapore 169856, Singapore; lim.kiat.hon@singhealth.com.sg; 6Department of Hematology, Singapore General Hospital, Singapore 169856, Singapore; william.hwang.y.k@singhealth.com.sg; 7National Cancer Centre Singapore, Singapore 169610, Singapore; 8Department of Surgery, Yong Loo Lin School of Medicine, National University of Singapore, Singapore 119228, Singapore; 9SingHealth Duke-NUS Medicine Academic Clinical Programme, Duke-NUS Medical School, Singapore 169857, Singapore

**Keywords:** cord lining mesenchymal stromal cells, systemic lupus erythematosus, myeloid cells, neutrophils, monocytes/macrophages

## Abstract

To determine the therapeutic efficacy of human umbilical cord lining mesenchymal stromal cells (CL-MSCs) (US Patent number 9,737,568) in lupus-prone MRL/lpr (Fas^lpr^) mice and elucidate its working mechanisms. A total of 4 doses of (20–25) × 10^6^ cells/kg of CL-MSCs was given to 16-week-old female Fas^lpr^ mice by intraperitoneal injection. Three subsequent doses were given on 17 weeks, 18 weeks, and 22 weeks, respectively. Six-week-old Fas^lpr^ mice were used as disease pre-onset controls. Mice were monitored for 10 weeks. Mouse kidney function was evaluated by examining complement component 3 (C3) deposition, urinary albumin-to-creatinine ratio (ACR), and lupus nephritis (LN) activity and chronicity. Working mechanisms were elucidated by flow cytometry, Luminex/ELISA (detection of anti-dsDNA and isotype antibodies), and RNA sequencing. CL-MSCs improved mice survival and kidney function by reducing LN activity and chronicity and lymphocyte infiltration over 10 weeks. CL-MSCs also reduced urinary ACR, renal complement C3 deposition, anti-dsDNA, and isotype antibodies that include IgA, IgG1, IgG2a, IgG2b, and IgM. Immune and cytokine profiling demonstrated that CL-MSCs dampened inflammation by suppressing splenic neutrophils and monocytes/macrophages, reducing plasma IL-6, IL-12, and CXCL1 and stabilizing plasma interferon-γ and TNF-α. RNA sequencing further showed that CL-MSCs mediated immunomodulation via concerted action of pro-proinflammatory cytokine-induced chemokines and production of nitric oxide in macrophages. CL-MSCs may provide a novel myeloid (neutrophils and monocytes/macrophages)-targeting therapy for SLE.

## 1. Introduction

Systemic lupus erythematosus (SLE) is a chronic multiorgan autoimmune disease that is potentially severe and even fatal. Breakdown of immune tolerance with the generation of antibodies to nucleic acids (NA), blood interferon (IFN) [1], and myeloid cells (mainly referring neutrophils, monocytes, and macrophages) are three hallmarks of SLE [2]. Garcia-Romo et al. and Caielli et al. demonstrated that netting neutrophils stimulated plasmacytoid dendritic cells (pDCs) to secret type I IFN directly [3] and further promoted the expansion of plasmablasts indirectly [4,5], suggesting the upstream effects of neutrophils in the pathogenesis of SLE.

Glucocorticosteroids and immunosuppressive regimens are the standard of care (SoC) for SLE [6]. Despite aggressive regimens, 30% to 60% of patients do not respond to these medications [7], and there are considerable side effects [8,9,10]. Because of partial expression of MHC class I, rare or no expression of MHC class II, and being negative for costimulatory molecules, such as CD80, CD86, and CD40, mesenchymal stromal cells (MSCs) are known to have low antigenicity [11]. Demonstrating a potent immunomodulatory effect through the induction of regulatory T and B cells (Tregs and Bregs) [12,13] and paracrine secretion [14,15,16,17,18,19,20,21,22,23,24,25], MSCs therapy has recently emerged as a potential alternative approach for the treatment of SLE. As of September 2022, MSCs have been used as therapeutics in 25 SLE clinical trials (reported on clinicaltrials.gov). However, MSCs have yet to be approved by the FDA for SLE treatment, mainly due to the inconsistencies in efficacy, which were exacerbated by the lack of consensus on the source, cell quantity, and frequency of MSC administration [26].

A review on the clinical use of MSCs to treat systemic SLE found that human bone marrow (BM)- and umbilical cord (UC)-derived MSCs are the two most common allogeneic cell sources used in clinical trials [26]. UC-MSCs are typically known to be derived mainly from the matrix or Wharton’s jelly (WJ) portion of the umbilical cords [27,28]. Otherwise, depending on the nature of isolation, these MSCs isolated would be heterogeneous, coming from both the WJ and the sub-amniotic lining membrane of the UC [29]. These umbilical cord lining (CL) MSCs are more recently described and found to be different from other extraembryonic tissue-derived MSCs derived from the placenta, WJ, and cord blood. CL-MSCs are isolated directly from the sub-amniotic layer [30] of the UC, which are homogenous with high ex vivo expansion potential [31] and hypo-immunogenic properties [32].

As there is currently no known study to report on the efficacy and working mechanism of human CL-MSCs for potential treatment of SLE, we aim to evaluate the effects of human CL-MSCs introduced intraperitoneally (IP) into lupus-prone MRL/lpr (Fas^lpr^) mice and elucidate its working mechanisms.

## 2. Results

### 2.1. CL-MSCs Were Able to Improve Mice Survival and Reduce Disease Activity through IP Administration

To demonstrate the therapeutic efficacy, we directly administered single and multiple doses of CL-MSCs to Fas^lpr^ lupus-prone mice by IP injection. Mice survival was improved from 25.0% (dPBS: 6/24 mice) to 50.0% (CL-MSCs (10 doses): 5/10 mice, *p* = 0.21) and 73.3% (CL-MSCs (4 doses): 11/15 mice, *p* < 0.05) at 10 weeks PT. There was no difference between mice treated with dPBS and a single dose of CL-MSCs (14.3%, 1/7 mice, *p* = 0.86) (Figure 1A). As four doses of CL-MSCs treatment demonstrated the optimal efficacy, all the following functional assays were only carried out with this four-dose treatment. Compared to dPBS-treated mice, CL-MSCs-treated mice exhibited a lower LN activity and chronicity at 10 weeks PT (*p* < 0.05 and *p* < 0.01) (Figure 1B). Furthermore, the LN chronicity was modulated back to the pre-onset level in all CL-MSCs-treated mice. This lower LN activity and chronicity was corroborated by reduced renal lymphocyte infiltration (Figure 1C). Taken together, the administering of CL-MSCs intraperitoneally improved mice survival and reduce disease activity.

### 2.2. Kidney Function Was Improved by CL-MSCs Treatment

To investigate the protective effect of CL-MSCs on kidney function, the levels of urinary albumin creatinine ratio (ACR) and renal complement C3 deposition were measured. Urinary ACR in CL-MSCs-treated mice (695.5 ± 619.1 mg/g) was significantly lower than dPBS-treated mice (12,660.1 ± 13,020.4 mg/g) at three weeks PT (*p* < 0.01) (Figure 2A). The number of renal complement C3 deposition in CL-MSCs-treated mice (8.3 ± 3.2 per field) was significantly lower than dPBS-treated mice (20.7 ± 12.3 per field) at three weeks PT (*p* < 0.05) (Figure 2B). Simultaneously, the intensity of fluorescent signals in CL-MSCs-treated mice was obviously weaker than dPBS-treated mice (Figure 2C). Taken together, the administering of CL-MSCs intraperitoneally improved mice kidney function.

### 2.3. Both Anti-dsDNA and Isotype Antibodies Were Reduced by CL-MSCs Treatment

To investigate the preventive effect of CL-MSCs on the breakdown of tolerance to NA, the levels of plasma anti-dsDNA antibodies were measured. Accumulation of anti-dsDNA (including IgA, IgG, and IgM) antibodies were reduced from (5.8 ± 4.1) × 10^6^ U/mL in dPBS-treated mice to (2.4 ± 2.5) × 10^6^ U/mL in CL-MSCs-treated mice at three weeks PT (*p* < 0.05) (Figure 3A). Concurrently, the levels of isotype antibodies (e.g., IgA, IgG1, IgG2a, IgG2b, and IgM) were significantly reduced in CL-MSCs-treated mice at three weeks PT, which were equivalent to the levels observed in six-week-old pre-onset mice (Figure 3B). Taken together, CL-MSCs were able to reduce both anti-dsDNA and isotype antibodies.

### 2.4. CL-MSCs Mediated Immunomodulation by Targeting Pro-Inflammatory Cytokine Secreting Monocytes/Macrophages and Neutrophils

To elucidate the possible mechanisms for our observations, we sought to identify immune cell subtypes and relevant cytokines that are targeted and modulated by CL-MSCs in vivo. We first examined how CL-MSCs treatment affected the immune cells in Fas^lpr^ mice by flow cytometry analysis. In 19-week-old Fas^lpr^ mice at 3 weeks PT, compared to dPBS-treated mice, we found that the numbers of splenic monocytes/macrophages (*p* < 0.05) and neutrophils (*p* < 0.001) were significantly lower in CL-MSCs-treated mice (Figure 4A). There was no difference on splenic B cells, cytotoxic T lymphocytes (CTL)s, T_H_1 and T_H_17 cells (Figure 4A). We then examined how CL-MSCs treatment affected the cytokine profile by Luminex assay. In 19-week-old Fas^lpr^ mice at 3 weeks PT, compared to dPBS-treated mice, CL-MSCs-treated mice had significantly lower plasma levels of pro-inflammatory cytokines, specifically IL-6, IL-12/IL-23 (p40), and KC (CXCL1) (Figure 4B). Although the plasma levels of TNF-α and IFN-γ in CL-MSCs-treated mice were not significantly lower than the levels observed in dPBS-treated mice, they were at the levels observed in six-week-old pre-onset mice. Together with the identification of target cells, we concluded that CL-MSCs treatment reduced inflammation in SLE by suppressing the proliferation of neutrophils, monocytes/macrophages, and therefore, their secretion of pro-inflammatory cytokines.

### 2.5. RNA Sequencing Confirmed the Immunomodulatory Effect of CL-MSCs in Murine SLE

To further confirm the findings in the above section, we examined the molecular changes caused by CL-MSCs using RNA-seq. Kidney tissues isolated from 26-week-old Fas^lpr^ mice that had been treated with four doses of CL-MSCs and 16-week-old Fas^lpr^ mice treated with 4 doses of dPBS (control) were compared.

To better understand how CL-MSCs modulated immune reactions, we used principal component analysis (PCA) to compare the molecular changes in CL-MSC-treated mice with control mice. CL-MSC-treated mice had significantly different gene expression from control mice (Figure 5A). Using the Ingenuity Pathway Analysis (IPA), we found that the most relevant pathways involved in CL-MSCs-mediated immunomodulation were the activation of T and B lymphocytes (an increase in the expression of genes *Cd79a*, *Cd79b*, *Fcgr3*, *Ikeke,* and *Pik3cd*), enhanced communication between innate and adaptive immune cells (an increase in the expression of genes *Il1b*, *Tnf*, *Ccr7*, *Trac,* and *Trbv*) as well as enhanced leukocyte extravasation (an increase in the expression of genes *Itgal*, *Itgb2*, *Mmp3*, *Mmp15*, *Selplg,* and *Vcam1*). We also found that Interleukin (IL)-15 signaling, a pathway associated with the activation and proliferation of T and natural killer (NK) cells, was activated with the increased expression of gene *Il2rg.* There was evidence of higher production of NO in the CL-MSC-treated group as the expression of gene *Nos2* was increased (Figure 5B–D).

## 3. Discussion

There are few existing therapeutic options for SLE, a disease characterized by both dysregulated innate and adaptive immunity. CL-MSCs, derived from the sub-amniotic umbilical cord lining membrane, are proving to be a promising source of stromal cells for displaying a more beneficial immunogenic profile and stronger overall immunosuppressive potential compared with BM-MSCs and other extraembryonic gestational tissue-derived MSCs [32,33]. Through a 10-week experiment, we have convincingly demonstrated that IP administration of CL-MSCs to Fas^lpr^ lupus-prone mice improved survival and reduced indices of disease activity. Immune and cytokine profiling demonstrated that IP MSC transplantation (MSCT) suppressed the proliferation of splenic neutrophils and monocytes/macrophages and blood proinflammatory cytokines (e.g., IL-6 and IL-12/23 (p40)). RNA sequencing further demonstrated that the immunomodulatory effects induced by IP MSCT were via the concerted action of pro-proinflammatory cytokine-induced chemokines and the production of NO in macrophages. As far as we are aware, this is the first report to demonstrate the myeloid-targeting therapy, not only monocytes/macrophages but also neutrophils, suggesting the promise of CL-MSCs in SLE therapy.

As we have observed an occurrence of lung emboli in a porcine model after IV infusion of Cl-MSCs (unpublished data), most likely due to the intrinsic presence of cell adhesion proteins and an absence of anti-cell adhesion proteins, we chose to administer CL-MSCs intraperitonially in the lupus-prone MRL/lpr (Fas^lpr^) mice to avoid any embolism in our current study. In recent years, there are increasing reports on the use of IP administration of MSCs [34,35] as an alternative to systemic IV administration, mainly to address concerns of pulmonary emboli and infarcts after IV injection [36,37,38]. IP-injected MSCs in direct comparison with IV administrations have been shown to have similar, if not better effects in many pre-clinical studies, such as experimental autoimmune encephalitis [39], mice with dextran sulfate sodium-induced colitis [40], and mice with zymosan-induced peritonitis [41]. In addition, IP injection of MSCs allows for a depot of exosome releasing cells in the abdominal cavity [35]; and more importantly, it permits the administration of a larger numbers of cells to address cell dosing limitation, which might be a critical factor for MSCs therapeutic success [42].

While IV administration of MSCs to treat severe and drug-refractory SLE has reached clinical trial stage [26], there is currently no known pre-clinical work to study the safety and efficacy of MSCs administered intraperitoneally. Using lupus-prone MRL/lpr (Fas^lpr^) mice, we found that IP injection of CL-MSCs at four timepoints increased survival of the treated mice by 48.3% compared to the dPBS-treated control mice. At the same time, there was improvement in the kidney function of these CL-MSC-treated mice as evident from the lower LN activity and chronicity, reduced renal complement C3 deposition, lower urinary ACR, as well as decreased lymphocyte infiltration at 10 weeks post treatment.

As for the immunological effects of IP-delivered CL-MSCs on lupus prone mice, we first demonstrated that CL-MSCs were able to mitigate the breakdown of tolerance to NA with the reduction in both anti-dsDNA and isotype antibodies in treated mice. Next, flow cytometry analysis of immune cells derived from the CL-MSC- and dPBS-treated mice revealed that CL-MSCs had a role in dampening inflammation by suppressing the number of splenic monocytes/macrophages and neutrophils. This reduced inflammation was further supported by data from the cytokine profiling of animals’ blood plasma where levels of pro-inflammatory cytokines consisting of IL-6, IL-12, and CXCL1 were found to be lower in the CL-MSC-treated group.

As compared to BM-MSCs in our previous study [43], CL-MSCs demonstrated better therapeutic efficacy with a higher survival rate (73% vs. 59%) and are more protective on kidney function with lower LN activity and chronicity, lower and more consistent urinary ACR, weaker renal complement C3 deposition signals, and less renal lymphocyte infiltration. Our current study on CL-MSCs showed that these cells control the development of murine lupus through myeloid suppression (neutrophils and monocytes/macrophages), while BM-MSCs through a concerted effect (promoting the proliferation of Tregs and Bregs; suppressing the activation and proliferation of T cells, B cells, NK cells, and DCs and promoting the polarization of macrophages from M1 to M2) [44]. Considering neutrophils are the upstream pathological players of SLE, we believe that a myeloid-targeting therapy using CL-MSCs will provide more effective therapy.

In our previous study, combination therapy with methylprednisolone and cyclophosphamide (CP) was used as a standard of care (SoC) control. In severe lupus mice (16-week-old mice), SoC lost its therapeutic effect and all mice died at approximately 58–60 days post-initiation of SoC treatment (within a week after the last dose of CP treatment (0.5 g per BSA, equivalent to 315 mg/kg)). This was similarly observed in Anton et al.’s study [45], in which 300 mg/kg CP was administered, and peak mortality (50%) occurred in DBA/2 mice at 54–68 days post-treatment accompanied by thymus involution. However, when we treated mice with CL-MSCs, most of these severe mice were able to survive through, suggesting that CL-MSCs therapy is safe especially in the vulnerable individuals.

MSC therapy has emerged as a potential approach for various immunological disorders. To date, studies indicate that induction of Tregs [12,13] and production of transforming growth factor-β [15], IL-10 [16], hepatocyte growth factor [17], indoleamine 2,3, dioxygenase [18,19], prostaglandins E2 [16,20], nitric oxide [14], TNF-α stimulated gene/protein 6 [21], heme oxygenase-1 [22,23], galectin-1 [24], and HLA-Gs [25] are attributable to the immunomodulatory function of MSCs [46]. Specifically, in Shi et al.’s study [14], they found that the immunosuppressive function of MSCs was elicited by IFN-γ and the concomitant presence of any of three other proinflammatory cytokines, TNF-α, IL-1α, or IL-1b. These cytokine combinations provoked the expression of high levels of several chemokines and inducible nitric oxide synthase (iNOS) by MSCs. Chemokines drove T cell migration into proximity with MSCs, where T cell responsiveness was suppressed by NO. Similarly, our RNA-seq data showed the upregulation of proinflammatory cytokine genes (*Il1b* and *Tnf*), chemokine and receptor genes (*Ccl9*, *Ccr7* and *Cxcr5*), and NO genes (*Nos2*) in post CL-MSCs treatment. Meanwhile, the activation of T and B lymphocytes, communication of innate and adaptive immune cells, and NO production in macrophages were increased as well. These data are strongly suggestive that the immunomodulatory function of our IP administered CL-MSCs is via the concerted action of chemokines and NO, which is consistent with Shi et al.’s report [14].

Our CL-MSCs, provided by CellResearch Corporation, have previously been characterized and found to have phenotypical characteristics of BM-MSCs in terms of STRO-1, CD73, CD44, and c-Kit expression as well as their capacity to differentiate into osteogenic, adipogenic, and chondrogenic lineages [47,48]. In our current mice study, we were able to inject up to an average cell dose of 22.5 × 10^6^ cells/kg at four timepoints, and this same regime can be translated to clinical trial use unlike the more matured bone-marrow (BM)-MSCs, which are currently restricted to 1 × 10^6^ cells/kg and with lower frequency in current clinical trials [26], most likely due to limitations in their growth capacity and numbers to treat an average-sized adult of 70 kg.

Overall, the use of IP injection of CL-MSCs was found to show great promise in our pre-clinical lupus model in terms of survival rate, kidney function, and immunomodulation effects that are equivalent, if not better, to what had previously been obtained using IV injection of BM-MSCs on the same pre-clinical model [49,50]. Further investigation is needed to understand the mechanisms of action and safety profile behind IP injection of MSCs as we build the case up toward clinical trials.

## 4. Materials and Methods

### 4.1. Mice

Fas^lpr^ mice were purchased from The Jackson Laboratory and bred in our animal facility. Only female mice (ICR and Fas^lpr^ mice) were used in this study, and they were housed in the SingHealth Experimental Medicine Centre under specific pathogen-free conditions. In this study, 6-week-old mice were considered to represent pre-onset SLE, while 16-week-old mice represented the severe stages of SLE. The animals were fed gamma-irradiated commercial rodent feed, and the facility used autoclaved water. The animals were given access to feed and water ad libitum. The environmental conditions were controlled with a temperature of 22 ± 2 °C, a relative humidity less than 70%, and a 12:12-h light: dark cycle. All cages were assembled with bedding and furniture before being autoclaved, the cages were changed every two weeks and/or whenever necessary as determined by the research personnel or animal care team.

### 4.2. Mice Treatment

Fas^lpr^ lupus-prone mice spontaneously developed lupus features, and the severity was associated with age. At 16 weeks, anti-double stranded DNA (anti-dsDNA) autoantibody increased approximately 100-fold compared to 6-week-old young mice. An exponential increase in serum autoantibody started from 6 weeks and plateaued at 15–17 weeks, indicating the full development of lupus.

To evaluate therapeutic efficacy, four arms of experiment were initiated: (i) dPBS negative control: 5 μL/g of body weight, IV, weekly, 10 doses in total; (ii) human CL-MSCs (single): provided by CellResearch Corp [47] ((20–25) × 10^6^ cells/kg, IP, 1 mL per mouse, resuspended in PTTe-1 culture medium, 1 dose; (iii) human CL-MSCs (10 doses): IP, weekly, 10 doses in total; and (iv) human CL-MSCs (4 doses): IP, 1 dose respectively on 16-week, 17-week, 18-week, and 22-week-old mice. Mice survival was monitored for 10 weeks post-initiation of treatment (PT).

As the optimal efficacy was achieved with 4 doses of CL-MSCs treatment, all the following functional and mechanistic studies only focused on this 4-dose-treatment strategy. For functional studies carried out in 19-week-old mice, the treatment was only given on 16th, 17th, and 18th week.

A separate batch of mice was used for complement component 3 (C3) deposition assay.

### 4.3. NIH Lupus Nephritis (LN) Activity and Chronicity Indices

Based on modified NIH activity and chronicity indices [51], a semi-quantitative grading system of pathologic features on kidney biopsies allows for monitoring response to treatment, and disease progression. The LN activity and chronicity were evaluated with formalin-fixed kidneys from 6-week-old pre-onset, 26-week-old dPBS-, and CL-MSC-treated Fas^lpr^ mice (4 doses of treatment on the 16th, 17th, 18th, and 22nd week) Briefly, indicators of disease activity include endocapillary hypercellularity, neutrophils, or karyorrhexis within glomerular capillary loops, fibrinoid necrosis, hyaline deposits, cellular or fibrocellular crescents, and interstitial inflammation. Crescents and fibrinoid necrosis are weighted twice as they have a worse impact on prognosis. Indicators of disease chronicity include the total percentage of global glomerulosclerosis, fibrous crescents, tubular atrophy, and interstitial fibrosis. The scoring is based on the percentage of glomeruli with each feature in the biopsy on a 0 to 3 scale, with a score of 0 = not present, 1 = <25% glomeruli, 2 = 25–50% glomeruli, and 3 indicating >50% glomeruli.

### 4.4. Renal Lymphocyte Infiltration

Overall, lymphocyte infiltration was qualitatively examined in formalin-fixed paraffin-embedded kidney tissues stained with H&E in 6-week-old pre-onset and 26-week-old dPBS- and CL-MSCs-treated mice (a total of 4 doses of treatment on the 16th, 17th, 18th, and 22nd week).

### 4.5. Albumin-to-Creatinine Ratio (ACR) Measurement

The degree of kidney damage in Fas^lpr^ mice was evaluated by measuring the urinary ACR. A total of 50 μL of urine was collected from 6-week-old pre-onset, 19-week-old dPBS-, and CL-MSC-treated Fas^lpr^ mice (treatment on the 16th, 17th, and 18th week) by bladder massage at 3 weeks post treatment (PT). Urinary albumin concentrations were detected using a mouse albumin ELISA kit (Abcam, Cambridge, UK). Creatinine levels were detected by Jaffe’s reaction using a creatinine colorimetric assay kit (Cayman, Ann Arbor, MI, USA). Both were performed according to the manufacturer’s instructions. ACR is the ratio of albumin concentration in milligrams to creatinine concentration in grams.

### 4.6. Renal Complement C3 Deposition Staining

The complement C3 deposition level was determined by immunofluorescent (IF) staining with antibodies against mouse C3 at a 1:50 dilution (Abcam) and FITC-conjugated rabbit anti-rat IgG at a 1:1000 dilution (Abcam) on snap-frozen kidney sections harvested from 6-week-old pre-onset and 19-week-old dPBS- and CL-MSCs-treated mice (treatment on the 16th, 17th, and 18th week).

### 4.7. Anti-dsDNA Antibody Analysis

The concentration of circulating anti-dsDNA antibody was determined with mouse anti-dsDNA Ig’s (Total A+G+M) ELISA kit (Alpha Diagnostic, Antonio, TX, USA). A total of 150–200 μL of mouse peripheral blood was collected from 6-week-old pre-onset and 19-week-old mice treated with dPBS and CL-MSCs (treatment on the 16th, 17th, and 18th week), respectively. Plasma samples were isolated from the peripheral blood by centrifugation at 800× *g* for 5 min, and the isolated mouse plasma samples were assayed according to manufacturer’s instructions.

### 4.8. Flow Cytometric Assay

Splenic cells isolated from 6-week-old pre-onset and 19-week-old mice treated with dPBS and CL-MSCs (treatment on the 16th, 17th, and 18th week), respectively, were subjected to flow cytometric assay. Briefly, the antibodies used for phenotyping were DAPI, mCD45-PE (30-F11), mCD3-PE-Cy7 (17A2), mCD8a-BV510 (53-6.7), mCD19-BB515 (1D3) (BD Biosciences, NJ, USA), mLy6G-APC (E50-2440) (eBioscience, San Diego, CA, USA), mCD11b-PerCP-Cy5.5 (M1/70) (BD Biosciences), mCD45-FITC (30F11) (Mlitenyi Biotec, Bergisch Gladbach, Germany), CD25-BV421 (PC61), FoxP3-PE (MF14), IFN-γ-PE-Cy7 (XMG1.2) (BD Biosciences), IL-4-APC (11B11), and IL-17A-BV510 (TC11-18H10.1). All antibodies were purchased from BioLegend unless otherwise specified.

A total of (2–5) × 10^5^ cells in 60 μL of dPBS was used to analyze surface marker expression by staining for 20 min at 4 °C, followed by flow cytometric analysis. For absolute counting, 10 μL of AccuCheck Counting Beads (Life Technologies, Carlsbad, CA, USA) was added to the flow tube after the final washing step.

For intracellular staining, mouse splenic cells were stained with surface antibodies first and then stimulated with 50 ng/mL phorbol myristate acetate (PMA) (Sigma-Aldrich, St. Louis, MO, USA) and 1 μg/mL ionomycin (Sigma-Aldrich) at 37 °C for 1 h and blocked with 1X brefeldin A (BioLegend, San Diego, CA, USA) at 37 °C overnight. Subsequently, the activated cells were fixed and permeabilized with a fixation/permeabilization solution (Miltenyi Biotec) for 30–45 min at 4 °C and stained with intracellular antibodies for 30 min at 4 °C for flow cytometric analysis. The assay was performed on Cyan ADP analyzer (Beckman Coulter, Brea, CA, USA) and data analyzed with Summit software 4.3.

### 4.9. Luminex Assay

The 19-week-old Fas^lpr^ mice treated with CL-MSCs and dPBS (treatment on the 16th, 17th, and 18th week) were used for a cytokine and an isotype antibody profiling study. The 6-week-old pre-onset mice were used as control. Mouse plasma cytokines/chemokines, and isotype antibodies were measured with a Luminex assay using a Bio-plex Pro^TM^ Mouse Cytokine 23-plex assay (Bio-Rad, Hercules, CA, USA) and MILLIPLEX^®^ Mouse Immunoglobulin Isotyping Magnetic Bead Panel—Isotyping Multiplex Assay (Merck, Rahway, NJ, USA) according to manufacturer’s instructions. This sandwich reaction occurred in wall-less DropArray (DA)-Bead plates (Curiox, Singapore, Singapore). By using this miniaturized plate, the reaction volume was reduced from 50 μL to 10 μL per reaction. In accordance with manufacturer’s instructions (https://youtu.be/P6ShRhE1GKE, accessed on 9 April 2020), the wall-less DA-Bead plate was blocked with dPBS containing 1% bovine serum albumin for 30 min. Ten microliters of mouse plasma and human plasma or serum with 4-fold dilutions were loaded onto the plate with preloaded magnet beads and incubated for 2 h. The reaction was further incubated with 10 μL detection antibody for 1 h and 5 μL streptavidin-phycoerythrine for 30 min. All incubations occurred on a 1-cm span orbital shaker at 385 rpm at room temperature. The plate was washed 3 times in the LT-MX washer between these three incubation steps. Plates were read on Luminex LX200 (Merck Millipore, Burlington, MA, USA) with Luminex xPONENT 3.1 software.

### 4.10. RNA Sequencing (RNA-Seq) and Data Analysis

The 26-week-old Fas^lpr^ mice treated with CL-MSCs (a total of 4 doses of treatment on the 16th, 17th, 18th, and 22nd week) and 16-week-old mice treated with dPBS (4 doses weekly) were used for RNA-sequencing. RNA samples were extracted from the mouse kidney by the standard TRIzol (Life Technologies, Carlsbad, CA, USA) RNA extraction method. The TruSeq Stranded mRNA LT Sample Prep Kit (Illumina, San Diego, CA, USA) with an Illumina NextSeq high 1 − 76 bp (multiplex) run type was used according to the manufacturer’s instructions. Raw reads were aligned to the mouse reference genome version GRCm38 gencode M23 (downloaded from https://www.gencodegenes.org/, accessed on 10 July 2021) using STAR aligner version 2.7.1a with default parameters. Read counts per gene were generated using the featureCounts function in the subread package version 2.0.0. Count per million (cpm) values were calculated using the cpm function of the edgeR R package. PCA was performed using the prcomp R function with log2cpm values. Hierarchical clustering of samples was performed using the hclust R function with (1-Pearson correlation) as the distance and the Ward method with log2cpm values. Differential expression analysis of CL-MSCs-treated versus dPBS-treated mice was performed using the edgeR package. Genes with an adjusted *p* value less than 0.05 were considered significantly differentially expressed genes (DEGs). Pathways significantly enriched for DEGs were identified using IPA 22.0 software.

### 4.11. Statistical Analysis

The results were expressed as the mean ± standard deviation (SD). Statistical tests were performed using Student’s *t*-test or one-way ANOVA with GraphPad Prism 9 (GraphPad, San Diego, CA, USA). A *p* value < 0.05 was considered statistically significant. Dunnett’s test was used to correct for multiple comparison.

## Figures and Tables

**Figure 1 ijms-24-00365-f001:**
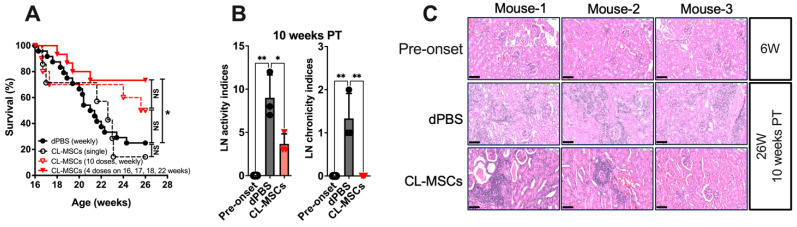
Intraperitoneal CL-MSCs were able to improve mice survival and reduce disease activity: (**A**) Survival curves of Fas^lpr^ mice treated with single and multiple doses of CL-MSCs. Twenty-four, 15, 7, and 10 mice were used in dPBS control, CL-MSCs (a total of 4 doses on 16-week-old, 17-week-old, 18-week-old, and 22-week-old mice), and CL-MSCs (single dose on 16-week-old mice) and CL-MSCs (a total of 10 doses weekly) groups. (**B**) Histopathological evaluation of LN activity and chronicity at 10 weeks PT. Three mice per group. (**C**) H&E staining. Kidney sections from each group at 10-week PT. Scale bar: 100 μm. Results were expressed as mean ± SD. For multiple comparison, the significance was defined as * *p* < 0.025; ** *p* < 0.005 when n = 2. One-way ANOVA Dunnett’s test was used for multiple comparison with 95% confidence interval.

**Figure 2 ijms-24-00365-f002:**
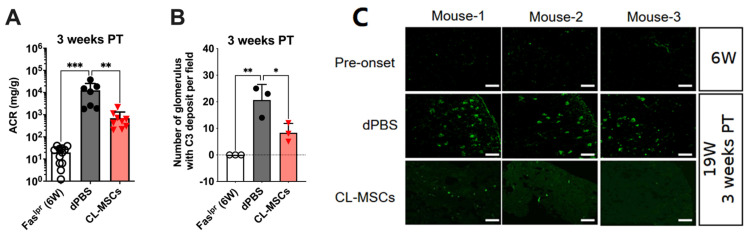
Kidney function was improved by CL-MSCs treatment: (**A**) Urinary ACR changes at three weeks PT. Twelve, 7, and 9 mice were used for 6-week-old pre-onset mice, dPBS control, and CL-MSCs. (**B**) The number of glomerulus with complement C3 deposition in the kidneys at three weeks PT (40-fold magnification). Three mice per group. (**C**) Representative image of complement C3 deposition. Green signals refer C3 deposition. Scale bar: 300 μm. Six-week-old Fas^lpr^ mice were used as a pre-onset control. Results were expressed as mean ± SD. For multiple comparison, the significance was defined as * *p* < 0.025; ** *p* < 0.005; *** *p* < 0.0005 when n = 2. One-way ANOVA Dunnett’s test was used for multiple comparison with 95% confidence interval.

**Figure 3 ijms-24-00365-f003:**
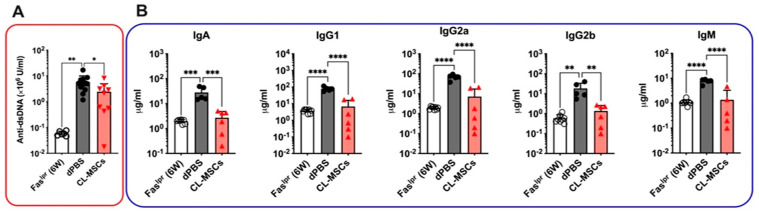
Both anti-dsDNA and isotype antibodies were reduced by CL-MSCs treatment: (**A**) Anti-dsDNA antibody changes in Fas^lpr^ mice treated with dPBS (n = 9) and CL-MSCs (n = 10) at three weeks PT. Six-week-old Fas^lpr^ mice (n = 12) were used as pre-onset control. (**B**) Isotype antibody changes in Fas^lpr^ mice treated with dPBS (n = 5) and CL-MSCs (n = 6) at three weeks PT. Six-week-old Fas^lpr^ mice (n = 8) were used as a pre-onset control. Results were expressed as mean ± SD. For multiple comparisons, the significance was defined as * *p* < 0.025; ** *p* < 0.005; *** *p* < 0.0005; **** *p* < 0.00005 when n = 2. One-way ANOVA Dunnett’s test was used for multiple comparison with 95% confidence interval.

**Figure 4 ijms-24-00365-f004:**
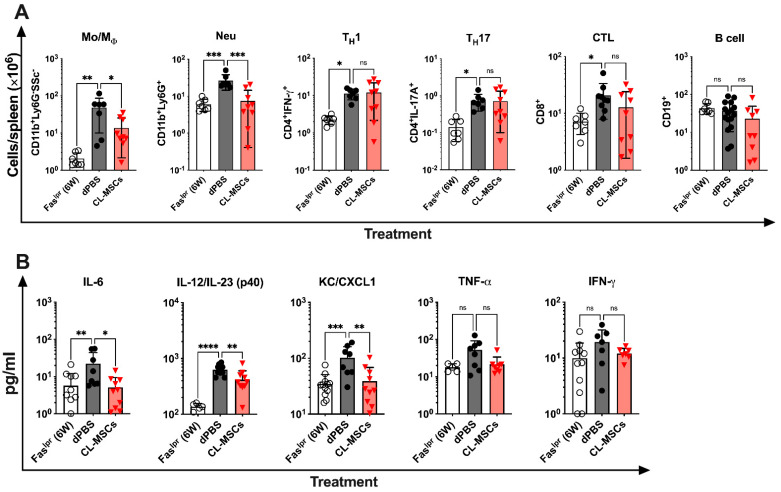
CL-MSCs mediated immunomodulation through targeting monocytes/macrophages and neutrophils and their pro-inflammatory cytokines: (**A**) The immune profile was analyzed on mouse spleen cells by flow cytometric analysis in six-week-old pre-onset Fas^lpr^ mice (n = 7) and at three weeks PT treated with dPBS (n = 7) and CL-MSCs (n = 9). (**B**) Cytokine profile of plasma samples was performed in six-week-old Fas^lpr^ mice (n = 9) and at three weeks PT treated with dPBS (n = 8) and CL-MSCs (n = 10) by Luminex assay. The results are expressed as the mean ± SD. For multiple comparisons, the significance was defined as * *p* < 0.025; ** *p* < 0.005; *** *p* < 0.0005; **** *p* < 0.00005 when n = 2. One-way ANOVA Dunnett’s test was used for multiple comparison with 95% confidence interval.

**Figure 5 ijms-24-00365-f005:**
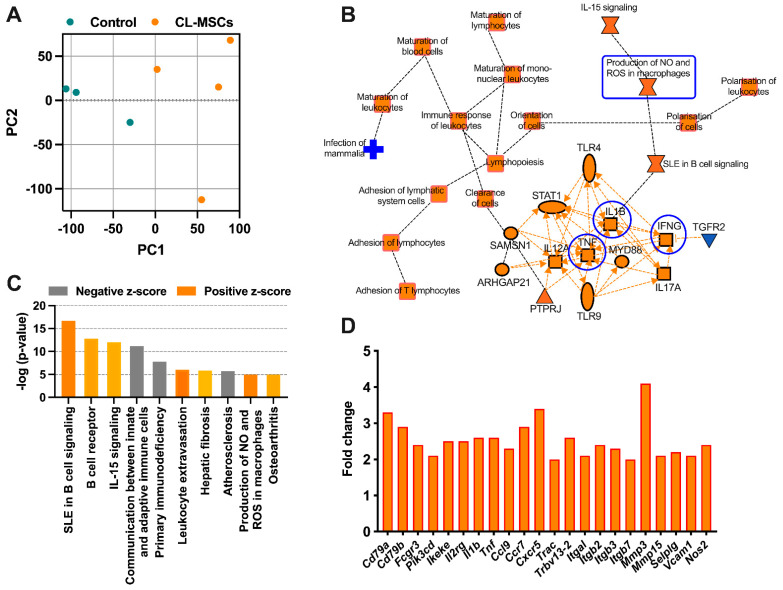
RNA sequencing confirmed CL-MSCs mediate immunomodulation via the concerted action of proinflammatory cytokine-induced chemokines and production of nitric oxide: (**A**) Principal component analysis (PCA) comparing 26-week-old Fas^lpr^ mice treated with four doses of CL-MSCs (n = 4) and 16-week-old Fas^lpr^ mice treated with four doses of dPBS (n = 3). Genes with an adjusted *p* value for false discovery rate (FDR) less than 0.05 were used for PCA. (**B**) Graphical summary of Ingenuity Pathway Analysis (IPA). (**C**) IPA showing the top 10 pathways identified by comparing CL-MSCs-treated mice versus control mice. A positive z-score represents activation of the pathway; a negative z-score represents inhibition of the pathway. (**D**) The fold change in typical genes involved in the top 10 pathways.

## Data Availability

There are no additional unpublished data. The RNA-seq data are available upon request.

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
