# Peer review of "Intraperitoneally Delivered Umbilical Cord Lining Mesenchymal Stromal Cells Improve Survival and Kidney Function in Murine Lupus via Myeloid Pathway Targeting"

_ijms, 2022, doi:10.3390/ijms24010365_

Round 1

Reviewer 1 Report (New Reviewer)

The manuscript entitled, “Intraperitoneally delivered umbilical cord lining mesenchymal stromal cells improve survival and kidney function in murine lupus via myeloid pathway targeting” targeted a novel strategy to improve the survival of kidney function in murine lupus. The study is well designed and properly executed; however, the authors needs to clarify my below concerns

1.      The authors mentioned about the dosage; which is 17, 18 and 22 weeks. How the authors have calculated and selected these specific weeks/days for MSCs administration.

2.      Secondly, the MSCs concentration which was 4 doses of (20-25) ×106 cells/kg of CL-MSCs. How the authors optimized the MSCs concentration to this specific point. Is there any trials for optimization of dosage have been conducted? If yes, then please mention it in Materials & Methods.

3.      As the authors use MSCs from human source (umbilical cord), then how they have tracked them inside mouse after IP injection and its localization/homing in different organs.

4.      Lastly, what was the retention percentage of MSCs in the damage organ (kidney)

Author Response

Thanks for your reviewing and please find the answers in the attached document.

Reviewer 2 Report (New Reviewer)

I commend the authors for their research on evaluating the efficacy of CL-MSCs in treating the kidney function in lupus disorder. The manuscript is well written. Introduction was brief, methods used were well explained. Results are well presented. Conclusions drawn from the results are logical. I suggest publishing the article in the current state.

Author Response

Thank you for your reviewing and support!

Round 2

Reviewer 1 Report (New Reviewer)

Dear Authors,

Thank you for your detail response to my comments.

This manuscript is a resubmission of an earlier submission. The following is a list of the peer review reports and author responses from that submission.

Round 1

Reviewer 1 Report

The manuscript “Intraperitoneally delivered umbilical cord lining mesenchymal stromal cells improve survival and kidney function in murine lupus via myeloid pathway targeting” by Chua et describes a study in MRL/lpr mice treated with human embryonic mesenchymal stem cells. While the topic is forefront in the field for the treatment of patients, there are some major concerns with the study design and analyses. The major study design concerns include:

1.     The lack of consistency in the groups being compared, and it was unclear how many mice were included in each set of analyses. The Methods section states that 24 control (PBS-treated) and 15 MSC-treated mice that were followed for 10 weeks post-treatment, which appear to all be included in the survival data in Figure 1. However, the authors do not state if a separate set of animals were used for the analyses of that were performed at 3 weeks post-treatment or how many mice were included. For these comparisons (Figs. 1-3), it is also unclear why the control group received 10 doses of PBS while the treatment group received only 4 doses of MSCs.

2.     For the RNA-seq analyses, untreated 16 week-old mice were compared to 26 week-old MSC-treated mice (10 week post-treatment). Thus, the authors are not comparing age-matched untreated versus MSC-treated mice. The effect of the MSC treatment was not compared to mice with a similar disease extent. It is more relevant to directly compare the 26 week-old MSC-treated to the 26 week-old control (PBS) mice.

3.     The authors fail to discuss the extensive published studies over the past 3-4 years conducted in both lupus mice and humans using MSCs to successfully treat disease, and how their study adds to what has already been demonstrated/shown:

a.     Xu et al., 2020. JASN 31 (doi.org/10.1681/ASN.2019050545)

b.     Lee et al., 2018. Lupus 27:1854-1859

c.     Deng et al., 2017. Ann Rheum Dis 76:1436–1439

d.     Liu et al., 2019. BioMed Research International

e.     Wang et al., 2018. Stem Cell Reports 13:933-941

Author Response

Dear Reviewer,

Thanks for your review and comments on our work.

I summarized the changes and answers to the attached file.

Thanks!

Reviewer 2 Report

I have reviewed the manuscript by Wen et al entitled Intraperitoneally Delivered UC–MSC Improve Survival and Kidney Function in Murine Lupus via Myeloid Pathway Targeting. To begin with the article is well organized with appropriate sections.  However, I do not believe that the introduction fully acknowledges the work that has been done in this field in the area of immune tolerance, lupus activity, and both pathophysiologic and genetic mechanisms that have been characterized using UC–MSC in the past. 

The introduction needs to be further developed, and the discussion should incorporate (and support or refute) prior observations in this field. Much has been written in the area of stem cell treatment for autoimmune diseases over the past 20- years; I am not sure what new ground is being broken in this particular work.  Perhaps the authors should specifically note  

observations which concur or refute prior information on both the pathophysiology and molecular observations as well as any NEW observations their work reveals.

While I am not very familiar with Principal Component Analysis, the data are convincing supporting the various postulated molecular pathways.  The methodology is clearly explained, the data is well supported as demonstrated in the figures. The paper is well written and easy to understand so long as the reader has some background in both lupus and involved immune mechanisms. Again, expansion of rationale for the study in the introduction would be useful.

I am also concerned about the absence in the discussion on the pros and cons of the methods and results.  For example, do all animal models respond similarly or are there differences in response to MSCs?  Along these lines I should note that trials in lupus patients have been undertaken with success certainly warranting continued research in this area. This should be noted. With attention to these issues I believe that this manuscript is worthy of publication.

Author Response

(The authors gave the same response as above.)

Round 2

Reviewer 1 Report

While the authors improved the Discussion and provided some clarifications surrounding the study desgin, there are still major concerns with the presented study. The study does not have a true control group. The MSC-treated mice were injected IP with cells in PTTE-1 culture medium, yet the "control" group was injected IV with dPBS. Moreover, it is still not entirely clear why some of the results are from mice treated with 3 doses of MSCs (such as C3 deposition) and other results are from mice treated with 4 doses of MSCs (such as lymphocyte infiltration). Did the authors not observe differences in all measures after 4 doses? There are too many inconsistencies with the study, thus direct comparisons and solid conclusions cannot be drawn.

Author Response

We appreciate Reviewer-1’s rigor on our experimental design. Actually, when we carried out this CL-MSCs study, we had another parallel study ongoing. In that study, we used the same mouse model (Faslpr mice) to demonstrate the therapeutic effect of chemokine CXCL5. These two works were submitted for publication around the same time. Because BM-MSCs and Standard of Care (SoC: methylprednisolone + cyclophosphamide) have been used as positive control in that study, I can’t reuse these data in this CL-MSCs study. But since CXCL5 work has been published in Oct 2022 (https://pubmed.ncbi.nlm.nih.gov/36240108/), now we can cite and compare with those positive controls directly. Hope Reviewer-1 can accept our work by using the IV dPBS as the negative control rather than IP dPBS.

In the discussion section, we cite our recently published work to show the difference, pros and cons within BM-MSCs, CL-MSCs and SoC on line 367-386:

As compared to BM-MSCs in our previous study45, CL-MSCs demonstrated better therapeutic efficacy with higher survival (73% vs 59%) and more protective on kidney function with lower LN activity and chronicity, lower and more consistent urinary ACR, weaker renal complement C3 deposition signals and less renal lymphocyte infiltration. Our current study on CL-MSCs showed that these cells control the development of murine lupus through myeloid suppression (neutrophils and monocytes/macrophages), while BM-MSCs through a concert effect (promoting the proliferation of Tregs and Bregs; suppressing the activation and proliferation of T cells, B cells, NK cells and DCs; and promoting the polarization of macrophages from M1 to M2)46. Considering neutrophils are the upstream pathological players of SLE, we believe that a myeloid-targeting therapy using CL-MSCs will provide more effective therapy.

In our previous study, combination therapy with methylprednisolone and cyclophosphamide (CP) was used as standard of care (SoC) control. In severe lupus mice (16-week-old mice), SoC lost its therapeutic effect and all mice died at approximately 58-60 days post-initiation of SoC treatment (within a week after the last dose of CP treatment (0.5 g per BSA, equivalent to 315 mg/kg)). This was similarly observed in Anton et al. study47, in which 300 mg/kg CP was administered, peak mortality (50%) occurred in DBA/2 mice at 54-68 days post-treatment accompanied by thymus involution. However, when we treated mice with CL-MSCs, most of these severe mice were able to survive through, suggesting that CL-MSCs therapy is safe especially in the vulnerable individuals.

The reason why we did not carry out the functional and mechanism studies on 26-week-old mice at 10 weeks post treatment was because only 25% of mice survived through if mice were just treated with dPBS. In another word, there was only around one mouse left if we started experiment with 5-6 mice. Hence, to achieve 8-10 data points per arm in the following experiments, we decided to run functional assay (urinary ACR, renal complement C3, anti-dsDNA and isotype antibodies), immune and cytokine profile studies on 19-week-old mice at 3 weeks PT as mouse blood, urine and immune cells need to be collected from alive mice.

To make the manuscript easy to read, I separated urinary ACR and renal complement C3 data measured at 3 weeks PT from mice survival and LN activity/chronicity and lymphocyte infiltration data measured at 10 weeks PT. Now they are under two sections.

3.1. CL-MSCs were able to improve mice survival and disease activity through IP administration

3.2. Kidney function was improved by CL-MSCs treatment
